# Field Experiments of Hyporheic Flow Affected by a Clay Lens

**Congcong Yao, Chengpeng Lu \*****, Wei Qin and Jiayun Lu**

College of Hydrology and Water Resources, Hohai University, Nanjing 210098, China
\* Correspondence: luchengpeng@hhu.edu.cn; Tel.: +86-258-378-7683

**Abstract:** As a typical water exchange of surface water and groundwater, hyporheic flow widely exists in streambeds and is significantly affected by the characteristics of sediment and surface water. In this study, a low-permeability clay lens was chosen to investigate the influence of the streambed heterogeneity on the hyporheic flow at a river section of the Xin'an River in Anhui Province, China. A 2D sand tank was constructed to simulate the natural streambed including a clay lens under different velocity of surface water velocity. Heat tracing was used in this study. In particular, six analytical solutions based on the amplitude ratio and phase shift of temperatures were applied to calculate the vertical hyporheic flux. The results of the six methods ranged from −102.4 to 137.5 m/day and showed significant spatial differences. In view of the robustness of the calculations and the rationality of the results, the amplitude ratio method was much better than the phase shift method. The existence of the clay lens had a significant influence on the hyporheic flow. Results shows that the vertical hyporheic flux in the model containing a clay lens was lower than that for the blank control, and the discrepancy of the hyporheic flow field on both sides of the lens was obvious. Several abnormal flow velocity zones appeared around the clay lens where the local hyporheic flow was suppressed or generally enhanced. The hyporheic flow fields at three test points had mild changes when the lens was placed in a shallow layer of the model, indicating that the surface water velocity only affect the hyporheic flow slightly. With the increasing depth of the clay lens, the patterns of the hyporheic flow fields at all test points were very close to those of the hyporheic flow field without a clay lens, indicating that the influence of surface water velocity on hyporheic flow appeared gradually. A probable maximum depth of the clay lens was 30 to 40 cm, which approached the bottom of the model and a clay lens buried lower than this maximum would not affect the hyporheic flow any more. Influenced by the clay lens, hyporheic flow was hindered or enhanced in different regions of streambed, which was also depended on the depth of lens and surface water velocity. Introducing a two-dimensional sand tank model in a field test is an attempt to simulate a natural streambed and may positively influence research on hyporheic flow.

**Keywords:** clay lens; heat tracing; field experiments; hyporheic flow

---

## 1. Introduction

As typical linear surface water bodies, rivers play an important role in many geological and ecological processes [1]. The hyporheic zone, which is the saturated zone alongside and beneath the streambed, is an active zone of surface water and groundwater interaction [2]. Driven by hydraulic head gradients, the stream water flows into and out of the hyporheic zone, which induces the exchange of mass and energy [3–6]. Owing to the unique characteristics of the hyporheic flow and biochemical environment, the hyporheic zone has significant effects on river–aquifer systems, including the solute transport [7–9], oxygen and nitrogen circulations [10–13], particle movement (e.g., colloid and heavy-metal ion) [14–17], and biological processes [18].

Hyporheic flow is mainly dominated by the patterns of the streambed, including its topography and structure [2,19,20]. The streambed topography, which changes the distribution of the hydraulic head, is one of the driving factors of the hyporheic flow [3,21–23]. The structure, which is the distribution of hydraulic conductivity (*K*), influences the hyporheic flow field and residence time [24–26]. Scouring and deposition lead to the redistribution of streambed sediment, resulting in the stratification which is ubiquitous in a natural streambed.

Fox et al. [27] constructed a 2D heterogeneous structure to evaluate the effects of different flow conditions on the hyporheic flow. Pryshlak et al. [28] chose a low-gradient stream with 300 different bimodal *K* fields to analyze the effects of heterogeneity in *K* on the hyporheic flux, residence time, and spatial pattern in a hyporheic zone. A variety of studies indicated that a heterogeneous streambed produces temporally and spatially variable hyporheic flows [29–32]. However, simulations of a heterogeneous streambed were random and unrealistic, and the condition of surface water were lack of consideration.

In this study, we examine the effects of a clay lens on the hyporheic flow. The clay lens is a typical low-permeable media and generally composed of peat and silt deposits. This impervious layer is widely distributed in streambeds, aquifers, and other strata, and mainly affects stream-groundwater interactions, mass migrations, and the exploitation of ground resources [33–35]. Studies about the relations between a clay lens and the hyporheic flow are rare, and there remain many challenges in determining the mechanism of hyporheic exchange in a heterogeneous streambed.

The application of heat as a tracer to determine hyporheic flow has become a general method in studies of hyporheic flows [36–39], groundwater recharge and discharge [40–44], and stream ecology [45,46]. As a novel and natural tracer, heat is pollution-free on the environment and can be monitored with low cost. Recently, the development of distributed temperature sensing technology has provided a means to obtain high-resolution temperature data, which is vital in delineating the hyporheic flow [47].

In this experimental study, we investigated the influence of a clay lens on the hyporheic flow in a natural stream. A 2D sand tank made of aluminum alloys and gauze was designed to simulate a heterogenous streambed for better quantitative and spatial control of the clay lens. This "artificial streambed" was placed to several sections of a river to experience different surface water velocities. As far as we know, there has been no research on hyporheic flow in a natural stream by using a sand tank. Comparing with the laboratory tests [48–50] and numerical simulations [51], this study reflects the characteristic of the mountain river in summer, making the hydraulic conditions of hyporheic flow more realistic. The vertical hyporheic flux (VHF) were calculated from raw temperature time series, which were monitored by temperature sensors buried in the sand tank. The specific objectives were (a) to analyze the characteristics of hyporheic flow in a streambed under the effect of a clay lens; and (b) demonstrate the relationship between the clay lens and hyporheic flow.

## 2. Site Description and Method

### 2.1. Site Description

The study sites were located in the Jinyuan River, a second-order river within the Xin'an River of Anhui Province, China. The length of the river was 373 km, and the area of catchment was 12,100 km$^2$. This catchment was bounded by Mount Huang and Yangtze River to the west and north, respectively. To the east and south, the catchment was bounded by Mount Tianmu and Mount Baiji.

The runoff of this catchment was influenced by the characteristics of the soil and the distribution of vegetation. The former was mainly related to the soil bulk density and porosity, and the latter depended on the land cover and land use. The soils in the study area included loam and clay. The vegetation coverage in high-altitude areas was larger than that in low-altitude areas, where part of the native vegetation had been destroyed owing to cultivation.

The lithology in the catchment was dominated by Proterozoic group strata covered with Quaternary sediment. Regional groundwater included pore water and fracture water in loose rock mass, and the pumpage capacity in a single well was up to 120–720 m$^3$/day. Owing to the large fluctuation of the surface landform and the poor storage ability of rock, groundwater recharged from precipitation was rapidly lateral discharged into the nearby valley in the form of a depression spring.

The distribution of the annual and monthly average precipitation was uneven and ranged from 913.9 mm to 2708.4 mm. Sixty percent of the annual precipitation occurred from April to July. The annual average evaporation was 851 mm, and the evaporation from May to August accounted for 50% of one year. Considering the spatial distribution, the runoff decreased from southwest to northeast, which was similar to the trend of precipitation. Most of the time during our field experiments the weather was sunny, and there was only a 7 mm rainfall event on 10 July.

The study river reach was nearly 100 m long, encompassing a section of tributary (Figure 1). The streambed consisted of fluvial sediments covering the bedrock where the average thickness of the sediment was 0.15 m. In this study, continuous observation of the water level and temperature in the streambed, river, and air were conducted over a ten-day period from 10 July 2018 to 20 July 2018.

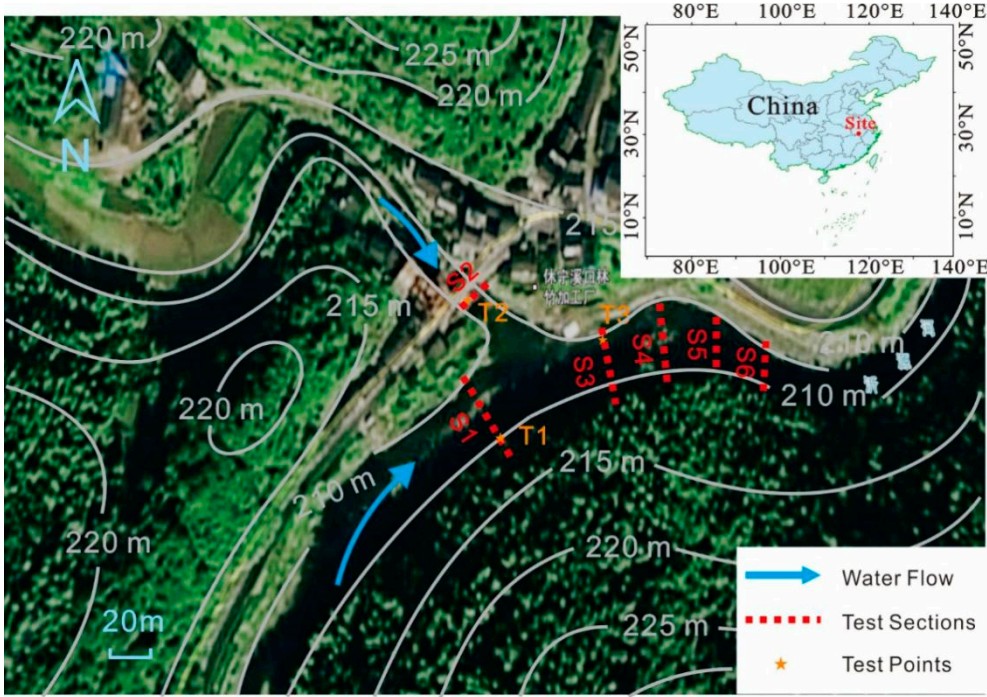

**Figure 1.** Study area of Jinyuan River and the hypsometric map of study area. Measurements of hydrologic factors, e.g., flow velocity, water depth, and section width, were taken at test sections S1, S2, S3, S4, S5, and S6. Field tests were conducted at test points T1, T2, and T3 located at S1, S2, and S3, respectively.

## 2.2. Hydraulic Characterization and Water Quality

Measurements of the hydraulic characterization of the stream and water quality in the study reach were conducted before the field test. The widths of six sections were determined with a digital level (DL201, South). A propeller-type current meter (LS10, Hydro-Bios) was used to measure the flow velocity of the surface water. The river discharge was calculated based on the weighted average velocity and water depth.

Sections S1 and S3 were located upstream and downstream of the main channel, respectively. The flow velocity, and water depth at S1 was much higher than those at S3, which meant that the upper reaches featured a higher drop with torrents and the flow in the downstream area become much slowly. Section S2 was located at tributary. Details of the hydraulic characterization are listed in Table 1.

**Table 1.** Hydraulic characterization of six sections and three test points.

| Section | Width, m | Maximum Depth, m | Average Flux, m³/s | Test Points | Distance to River Bank, m | Flow Velocity, m/s |
|---------|----------|------------------|--------------------|-------------|---------------------------|--------------------|
| S1 | 24.0 | 0.95 | 6.46 | T1 | 5.8 | 0.46 |
| S2 | 9.7 | 0.62 | 1.47 | T2 | 3.9 | 0.37 |
| S3 | 27.6 | 0.55 | 1.93 | T3 | 5.0 | 0.15 |
| S4 | 27.0 | 0.72 | 3.82 | | | |
| S5 | 24.6 | 0.82 | 4.38 | | | |
| S6 | 22.4 | 0.92 | 4.52 | | | |

In addition to the hydraulic characterization, several kinds of water samples from the stream, well, and soil were collected and measured in situ (Table 2). T1, T2, and T3 were selected as three sampling sites for surface water. Additionally, soil water at three different depths was collected at a natural overland flow site. In comparing the water quality, the stream water sample was clear and had lower solute concentrations, while the groundwater and soil water were muddy and alkaline, which meant that quality of the groundwater and soil water was relatively worse than surface water.

**Table 2.** Water quality of stream, groundwater, and soil water. A, B, and C were stream water samples from T1, T2, and T3. D was groundwater sample from well. E, F, and G were soil water samples at depths of 10 cm, 20 cm, and 30 cm, respectively.

| Sample | Temperature (°C) | EC (μs) | Salinity (ppt) | TDS (mg/L) | pH |
|--------|------------------|---------|----------------|------------|------|
| A | 25.3 | 43.3 | 0.02 | 30.8 | 7.46 |
| B | 23.9 | 30.6 | 0.01 | 21.8 | 7.48 |
| C | 24.5 | 30.3 | 0.01 | 21.8 | 7.49 |
| D | 26.1 | 933 | 0.46 | 662 | 9.43 |
| E | 28.1 | 148.5 | 0.07 | 105 | 8.55 |
| F | 28.1 | 192 | 0.09 | 137 | 8.22 |
| G | 27.3 | 364 | 0.18 | 258 | 9.28 |

## *2.3. Analysis Methods of Vertical Hyporheic Flux*

Time series of temperature in stream, air, and groundwater have significantly different trends. This makes it possible to reflect the exchange of water and mass between the stream and groundwater. Hatch et al. [52] proposed an analytical model for determining the streambed seepage using time series thermal data. The analytical model defined the relationships between the seepage and the amplitude ratio ($A_r$) or the lag time ($\triangle\phi$) of the temperature signal. On this basis, Luce et al. [53], Keery et al. [54], and McCallum et al. [55] modified and proposed new solution methods. All of these heat tracing models are based on a one-dimensional conduction–advection–dispersion equation [56–58]:

$$\frac{\partial T}{\partial t} = \kappa_e \frac{\partial^2 T}{\partial z^2} - \frac{n v_f}{\gamma} \frac{\partial T}{\partial z} \tag{1}$$

where $T$ is the temperature (varies with time $t$ and depth $z$); $\kappa_e$ is the effective thermal diffusivity; $\gamma = \rho c / \rho_f c_f$, the ratio of the heat capacity of the streambed to the fluid ($\rho_f c_f$ is the heat capacity of the fluid, and $\rho c$ is the heat capacity of the saturated sediment–fluid system); $n$ is the porosity, and $v_f$ is the vertical fluid velocity (positive = upwards flow) [52].

The solution to equation (1), given periodic variations in temperature at the top of the half-space, is modified from Goto et al. [59] and Stallman [57], and is rearranged to solve for the velocity of a thermal front as a function of amplitude and phase relations (Equations (2) and (3), or $v_{\mathrm{Ar}}$ and $v_{\triangle\Phi}$, respectively):

$$v_{Ar} = \frac{2\kappa_e}{\Delta z} \ln A_r + \sqrt{\frac{\alpha + v^2}{2}} \tag{2}$$

$$v_{\Delta\Phi} = \sqrt{\alpha - 2\left(\frac{\Delta\Phi 4\pi\kappa_e}{P\Delta z}\right)^2}$$ (3)

where $A$ is the amplitude of the temperature variations at the upper boundary, $P$ is the period of temperature variations ($P = 1/f$, where $f$ is the frequency), and $\alpha = \sqrt{v^4 + (8\pi \times \kappa_e/P)^2}$. Here, $A_r$ is the ratio of the amplitude ($A_r = A_d/A_s$), and $\triangle\Phi$ is the phase-shift variation between measurement points at different depths. The fluid velocities can be calculated from the thermal front velocities based on the relationships $v_{f,Ar} = v_{Ar}\,\gamma$ and $v_{f,\,\triangle\Phi} = v_{\triangle\Phi}\gamma$. In addition, the effective thermal diffusivity $\kappa_e$ is defined as:

$$\kappa_e = \frac{\lambda_e}{\rho c} = \frac{\lambda_0}{\rho c} + \beta|v_f|$$ (4)

where $\lambda_e$ is the effective thermal conductivity, $\lambda_0$ is the baseline thermal conductivity (in the absence of fluid flow), and $\beta$ is the thermal dispersivity.

Gordon et al. [60] developed a computer program named VFLUX to calculate the vertical water flux (in short, VHF). The analytical solutions were provided by Hatch et al. [52] to calculate the time series of the flux using either the amplitude ratio (the method HatchA) or phase shift (the method HatchP) of diurnal signals between two temperature time series.

On this basis, Keery et al. [54] improved and proposed a new analytical solution that was based on the amplitude ratio (the method KeeryA) or phase shift (the method KeeryP) without thermal dispersivity. In addition, Luce et al. [53] and McCallum et al. [55] presented novel analytical methods utilizing the amplitude and phase information, respectively (herein referred to as the method of McCallum and Luce). As the influence of nonideal field conditions on the amplitude ratio and phase-shift models has not been investigated thoroughly, Irvine et al. [61] included and evaluated many factors (e.g., non-sinusoidal temperature signals, unsteady flows, and multidimensional flows) within an updated version of VFLUX2. This model is composed of all the above methods. The sediment and thermal properties should be input before the program is run.

Raw temperature time series are recorded by sensors buried in the streambed. Dynamic Harmonic Regression (DHR) is used to isolate the diurnal signals in order to extract the amplitude and phase angle information. DHR is a method for nonstationary time series analysis that is particularly useful for extracting harmonic signals from dynamic environmental systems [62]. Here, DHR can produce time-varying apparent amplitude and phase coefficients for a time series at a user-specified frequency (10 min in this study).

Thermal parameters are considered constant in spatial and temporal distribution because of the relatively thermal homogeneity of streambed sediment. The values are determined through field observation, referring to several published studies [38,63,64].

### 2.4. Design of Temperature Sensors and Field Test

To simulate a natural streambed, we constructed a 2D sand-channel model in which the frame was welded together by aluminum alloy and wire netting wrapped in gauze, and was paved internally in order to fill the experimental material (sand, clay lens, and temperature sensors) and allow for water flow. The size of the model was 50 cm in length, 10 cm in width, and 50 cm in height (Figure 2a). The size of the clay lens was 10 cm in length, 5 cm in width, and 10 cm in height. Since the width of model and the height of lens were equal, when the lens was placed horizontally it attached the two sides of model (Figure 2b). Sensors were buried in the model to record the temperature distribution of this "artificial streambed". The interval between sensors in the horizontal and vertical directions was 10 cm and refined to 5 cm around the clay lens (from x = 10 cm and z = 10 cm to x = 40 cm and z = 40 cm, Figure 2b).

Based on the orthogonal test method, three test points (T1, T2, and T3) were chosen to test the influences of the surface water velocity. At each point, the depth of the clay lens was changed from 20 cm to 40 cm (Figure 2b). Twelve test scenarios are listed in Table 3. In addition to the "artificial

streambed", five temperature sensors were fixed on PVC tubes at 2-cm intervals. The tubes were then inserted into the natural streambed to record the temperatures at different depths from 2 cm to 10 cm (Figure 2c). The water levels, air pressure, and temperature were monitored by a level meter (Solinst 3001 Levelogger).

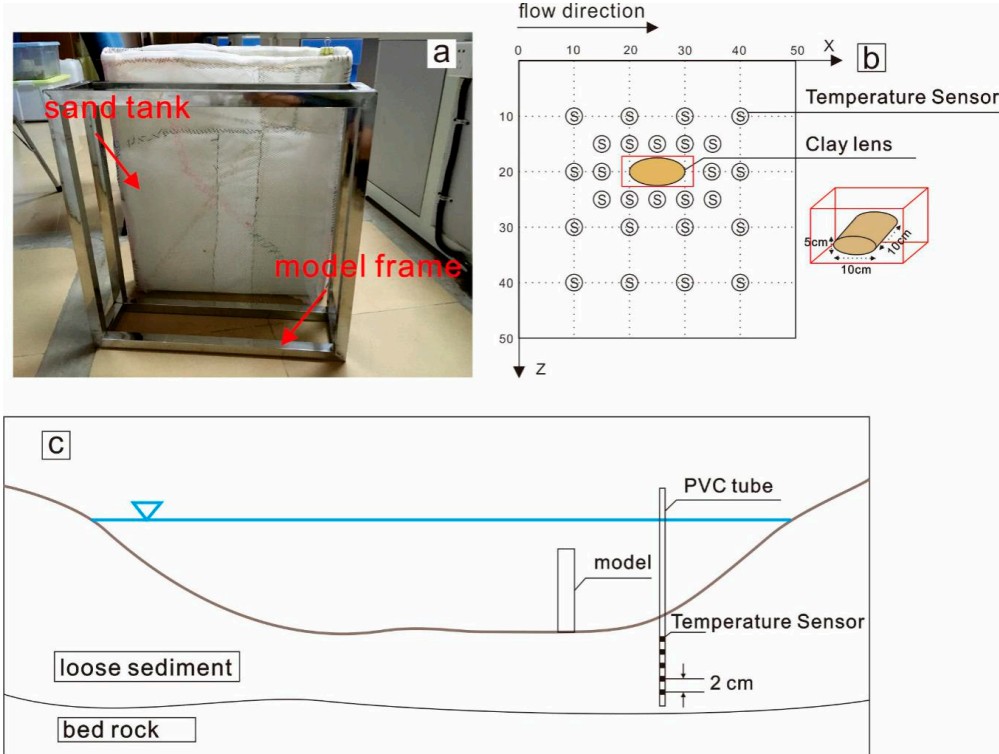

**Figure 2.** (**a**) Photo of model frame and sand tank before completion of construction. (**b**) Temperature sensors and clay lens distribution when depth of lens is 20 cm. Relative position when depth is 30 cm and 40 cm is consistent. (**c**) Sketch map of a cross-section of the study site.

**Table 3.** Summary of test scenarios, depths, and test points. N1–N12 are test numbers.

| Test Points | No Lens | Depth (m) | | |
|:---:|:---:|:---:|:---:|:---:|
| | | 0.2 | 0.3 | 0.4 |
| T1 | N1 | N2 | N3 | N4 |
| T2 | N5 | N6 | N7 | N8 |
| T3 | N9 | N10 | N11 | N12 |

In the test period, the model was placed at three test points (T1, T2, and T3), and the depth of the lens changed from 0.2 m to 0.4 m (Table 3). The temperature sensor was an iButton Thermochrons (model DS1922L) manufactured by Dallas Semiconductor. It had a 17.35-mm diameter and 6-mm thickness and was quite wearproof, making it suitable for fieldwork. Users can set different parameters according to their work needs. In this study, the accuracy, resolution, and response time of the sensors were 0.5 °C, 0.0625 °C, and 5 min.

## 3. Results

### 3.1. Temperature and Water Level of Natural Streambed

Figure 3 shows the temperature fluctuation in the air and natural streambed at different depths for T1, T2, and T3 from July 11 to July 20. Temperature changes with time formed a quasi-sine wave and change smoothly. Some small irregular fluctuations appeared during the test period, possibly

because of changes in the solar radiation and wind. At the end of each test, the temperature exhibited a rapid increase because the sensors were moved out from the stream.

Compared with the streambed temperature, the amplitude of the air temperature was significantly larger. The maximum and minimum of the air temperature were 30.9–35.7 °C and 21.8–22.9 °C, respectively, while in the streambed they were 26.4–30.7 °C and 22.7–25.2 °C. The air-temperature curve fluctuated more sharply, which were distributed in a serrated pattern owing to the great difference in heat capacity between air and water (the air temperature is more easily varied by solar radiation).

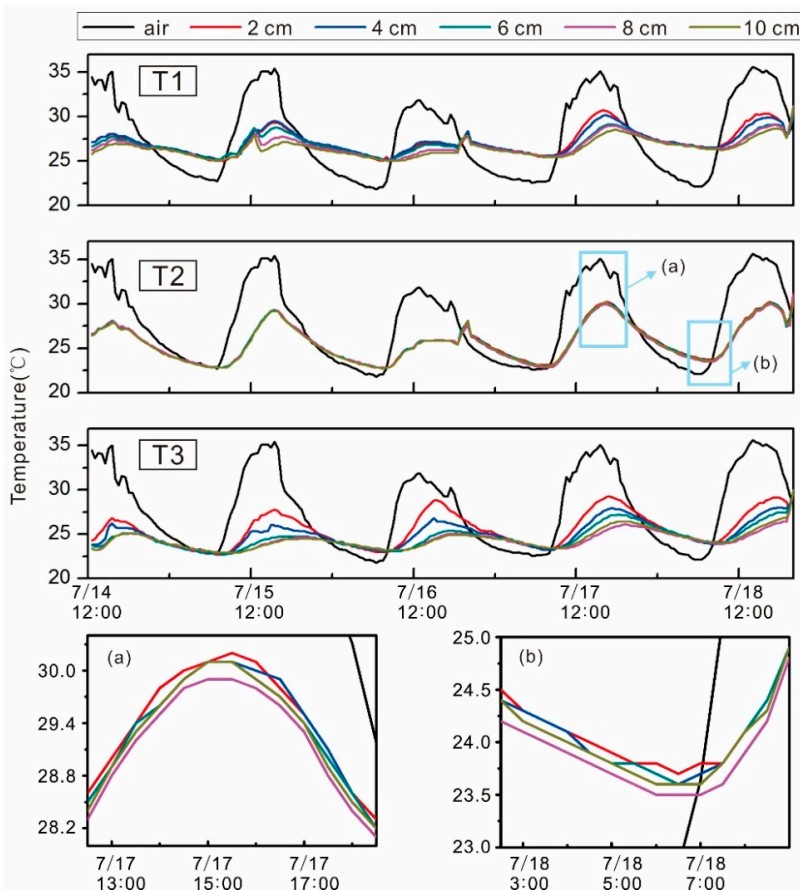

**Figure 3.** Variations of temperature and water level with time. Two periods of temperature were chosen to partially enlarge the curve (**a**,**b**).

*3.2. Comparison of Six Methods in VFLUX2*

Table 4 lists the values of the parameters used in VFLUX2. The data was selected from several similar field tests [43,65,66]. Figure 4 showed the mean and coefficient of variation (Cv) of the VHF calculated by six analytic methods. The six methods had significant differences that the order of VHF ranged from $10^{-2}$ to $10^{2}$ m/day. Hence, the results of HatchA and KeeryA were enlarged and shown in Figure 4.

**Table 4.** Physical properties and thermal parameters used in VFLUX2.

| | Porosity | Density (kg m$^{-3}$) | Volumetric Heat Capacity (cm$^3$ C) | Thermal Conductivity (s cm °C) | Dispersivity |
|---|---|---|---|---|---|
| Water | | 1000 | 0.5 | | |
| Sand | 0.372 | 2580 | 1 | | |
| Saturated Sediment | | | | 0.0045 | 0.001 |

Specifically, the VHF as calculated by the two amplitude methods (HatchA and KeeryA) were extremely close. The means of HatchA and KeeryA at T1, T2, and T3 were 0.142, 2.169, and −0.190 m/day, and 0.142, 2.123, and −0.184 m/day, respectively. The variation in the VHF was apparent only at T2, where Cv of HatchA and KeeryA were 0.023 and 0.222, respectively. Since the variation was too small, it was difficult to find the fine distinctions in Figure 4. In addition, calculations of HatchP and KeeryP at T2 were missing, i.e., only included the results at T1 and T3; this was mainly caused by the limitation of the phase shift methods.

Based on the raw temperature data and hydrogeological investigations, the streambed in the study reach was bedrock covered with loose sediment whose average thickness was around 15 to 20 cm. Thus, the interval between the two sensors was only 2 cm (as mentioned in Section 2.4). Hence, the heat transport was sometimes abnormal at T3. The temperatures at deeper points reached the maximum earlier than those at shallower points. That is, the phase shift was negative, and thus the VHF could not be calculated by the phase shift methods (HatchP and KeeryP).

The methods proposed by Hatch and Keery used a single information (amplitude ratio or phase shift), while the methods proposed by Luce and McCallum combined the amplitude ratio with the phase shift. However, calculations of these "synthesis" methods at three test points were quite different. The means of Luce and McCallum at T1, T2, and T3 were −0.853, −38.892, and −0.433 m/day, and −0.432, −38.892, and −0.403 m/day, respectively. In addition to different points, it was apparent that the variation in the VHF at the same point at different depths indicated that Cv from Luce and McCallum were much higher among the six methods.

From a comparison of the six methods, the methods based on the amplitude ratio and phase shift have a broader scope and higher robustness, which can be used in most cases. The phenomenon of missing calculations is insignificant. However, the results from Luce and McCallum are more easily affected by environmental factors and exhibit great fluctuations, which may lead to lower reliability and even errors. Additionally, the phase shift method has a higher requirement for raw temperature data. In this study, the changes in the amplitude ratio met the test demands. Therefore, the method of HatchA was used to calculate the VHF.

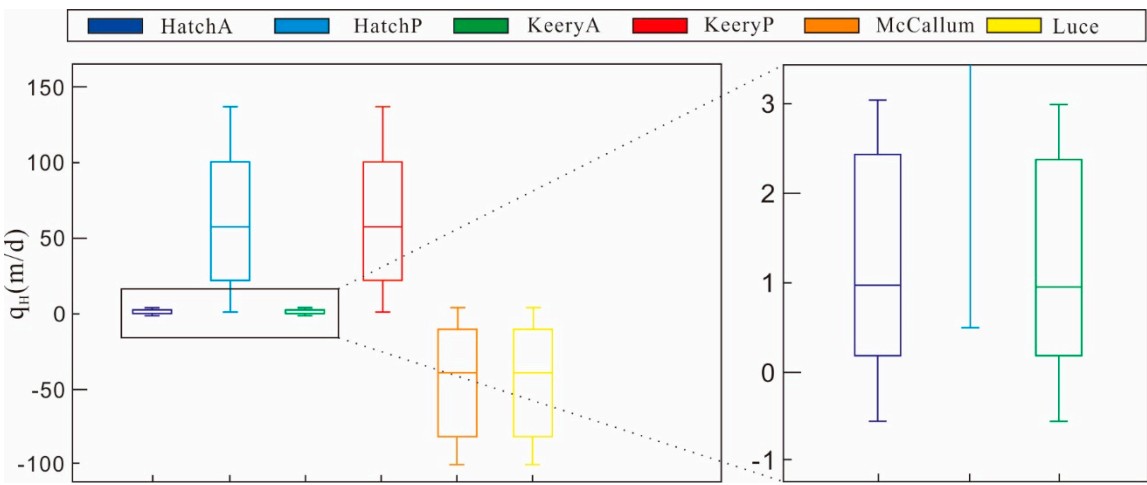

**Figure 4.** Box plots of hyporheic fluxes by six analytic methods.

### 3.3. Range and Spatial Distribution of Vertical Hyporheic Flux

Figure 5 shows the ranges of the VHF calculated by HatchA at depths from 0.03 m to 0.09 m, which were determined by the depths of the sensors and the rule of slide windows. The absolute value of the VHF at T1 was lowest at the three test points where the mean ranged from −0.178 to 0.247 m/day and the median ranged from −0.126 to 0.227 m/day. The VHF was close to 0, indicating that the hyporheic flow was relatively weak. It is noted that the positive and negative signs represent the direction of the VHF, i.e., a positive value indicates water flowing from the stream to the streambed.

The VHF at T2 ranged from 1.401 to 2.952 m/day, and decreased with depth except at 6 cm. This was an abnormal value of up to 6.790 m/day. T2 was located at a tributary where the particles of streambed sediment were coarser, including a few types of gravel. Thus, the hydraulic conductivity of the streambed at T2 was higher. Hyporheic flow was greater in the shallow layer and weakened in the deep layer. The direction of hyporheic flow at T3 included upwelling and downwelling flows, and this situation was similar as that of T1. However, the VHF in the shallow layer exhibited upwelling, which is different with T1. With an increase in depth, the VHF weakened gradually and transformed to a downwelling at the depth of 7 cm. The closer to the bedrock, the higher the VHF.

The VHF in a natural streambed is intimately linked to the sediments of the streambed and the surface water velocity. In general, at a site with deep water depth and high surface water velocity, the streambed has always been scoured and moved, and thus the VHF is suppressed significantly by the horizontal water flow. By contrast, in a low-velocity zone, the condition of surface water velocity is more beneficial to hyporheic flow in the vertical direction. In addition, as a special kind of porous flow, the lithology and construction of sediment obviously modifies the spatial distribution of hydraulic conductivity and then influences the hyporheic flux and residence time of the hyporheic flow.

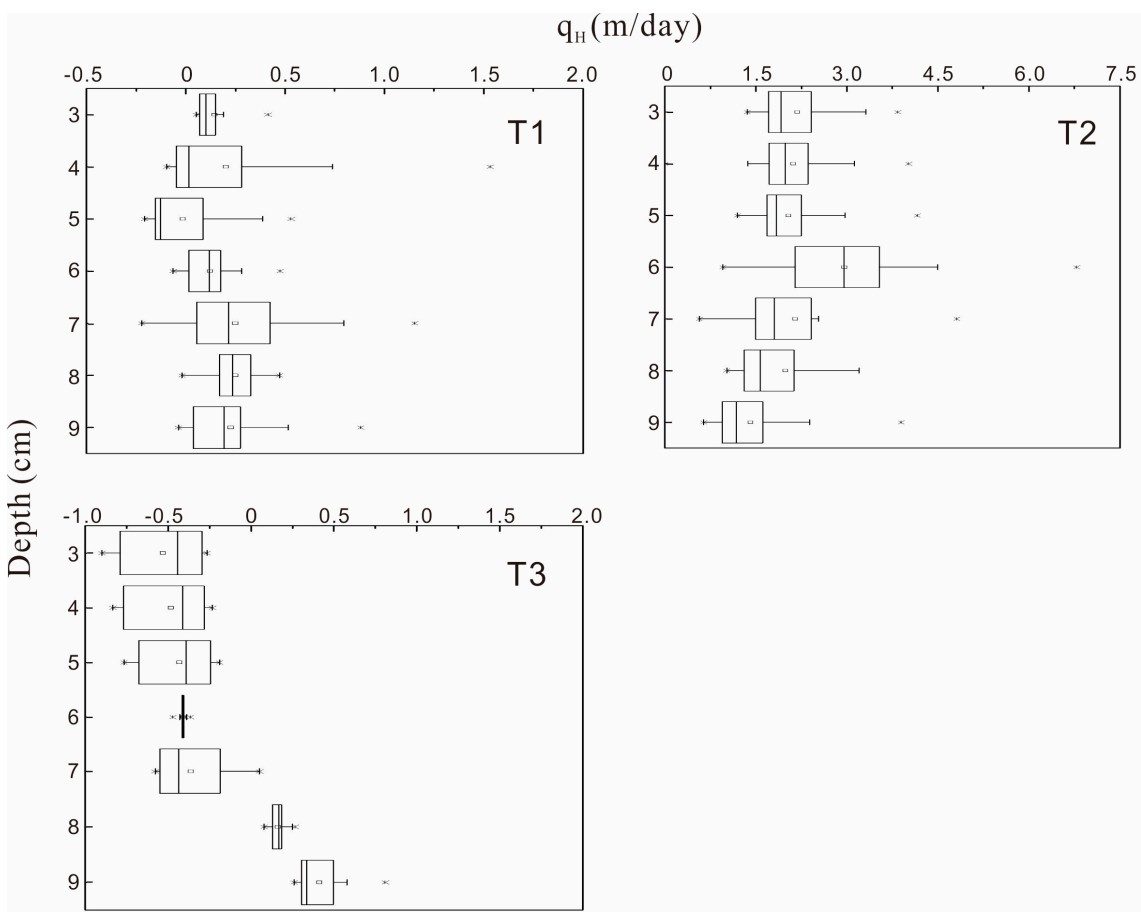

**Figure 5.** Statistics of vertical hyporheic flux (VHF) at different depths for T1, T2, and T3. Number of VHFs depended on frequency of temperature monitoring. Here, VHF was calculated at the interval of 30 min.

The field experiments were embedded in the natural environment in order to analyze the influence of the clay lens and surface water velocity on the VHF. Each experiment lasted more than 6 h to ensure the reliability of the raw temperature data and that the amplitude and phase signal could be extracted correctly. Figure 6 shows the spatial distribution of VHF at the artificial streambed without clay lens at

T1, T2, and T3, indicating changes in the VHF with time. The pattern of the surface water velocity at each point probably does not change a lot owing to the short test period for each experiment.

The VHF field varied slightly during the test period. The area of the high-velocity zone increased slightly and was mainly distributed around the deep zone (>25 cm). Focusing on the characteristics of the three test points, the VHF was lower in the shallow zone (<15 cm) and had small changes spatially. Several maximums ranging from 2.24 to 3.14 m/day appeared in the deep zone (>25 cm). In general, the VHF fields at T1 and T3 were similar probably because these two points were located in the main stream. Specifically, the VHF at T1 ranged from 0.73 to 3.14 m/day and from 0.97 to 2.57 m/day at T3. At T2, the maximum VHF occurred at around 35 cm in depth, which was deeper than those at T1 and T3. The topography of the tributary streambed had an obvious undulation, producing a multistage terrace, hydraulic drop, and hydraulic jump. This may be why the VHF field at T2 was different with those of the other two test points.

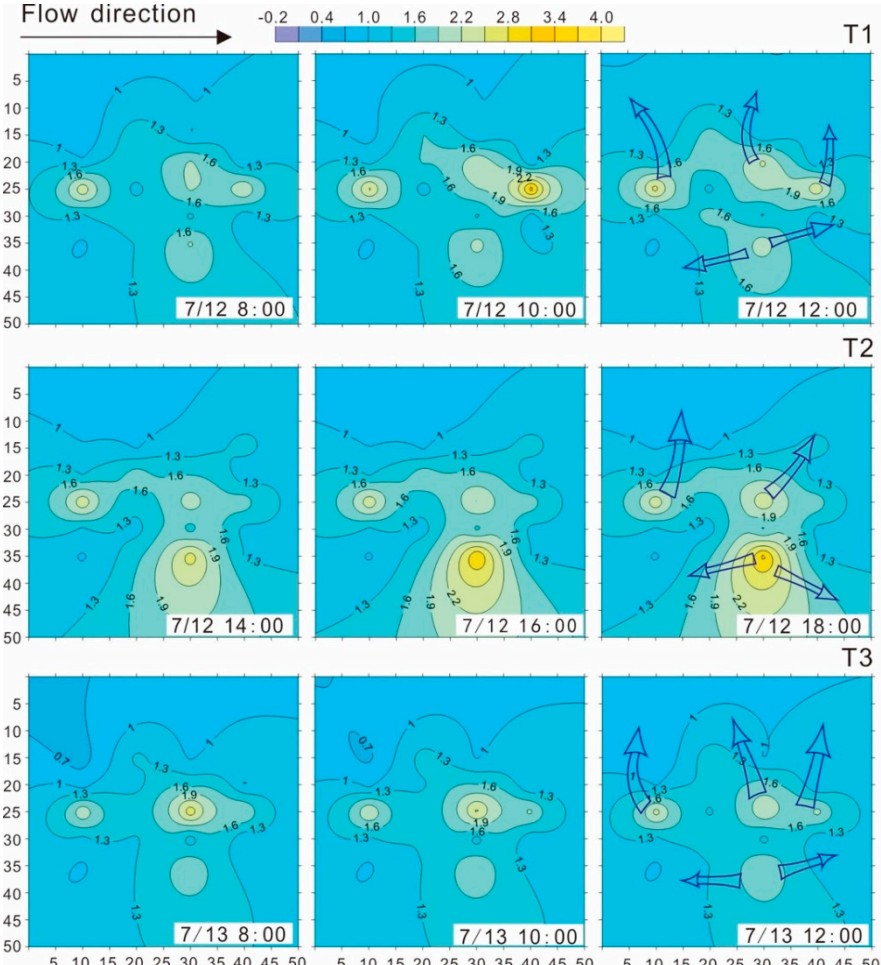

**Figure 6.** VHF field at T1, T2, and T3, calculated based on spatial temperature series. Contours were generated by the kriging method. The field experiments was carried out from 6:00 to 18:00 every day, and the interval between two VHF fields was 2 h. Surface water flowed from left to right.

## 4. Discussion

### 4.1. Influencing Factors on Hyporheic Flux

VHF fields influenced by clay lens depths and surface water velocity. Different from the blank control, the existence of the clay lens divided the artificial streambed into two parts, including the upper area and the downside area of the clay lens. Several pairs of sensors were separated at the two

sides of the clay lens. As a kind of low-permeable sediment, the VHF located at the point covered with a clay lens was blocked by the clay and the VHF value should be extremely low, which was not consistent with the theoretical value calculated by these "separated" sensor pairs. Therefore, the VHF in the covered zone should be excluded in the spatial analyses.

Figure 7 shows the influences of the clay lens and surface water velocity on the VHF. Jin et al. [17] found that the clogged layer with fine particles could reduce the hyporheic flow. In most scenarios with a clay lens, the VHF was generally less than the value for the blank control. The existence of a clay lens suppressed the vertical hyporheic flow to some extent. It was clear that the clay lens affected the local VHF, such as at 20 cm in depth, where the contours were distorted and moved toward the lens. In the region of the clay lens, the contours became more concentrated, meaning that the VHF dropped rapidly. The VHF often appeared below the lens (30 cm in depth) and above the lens (40 cm in depth at T1 and T3). Gomez-Velez et al. [34] posited that low-permeability layers induce hydrodynamic sequestration owing to the relocation and emergence of new stagnation zones.

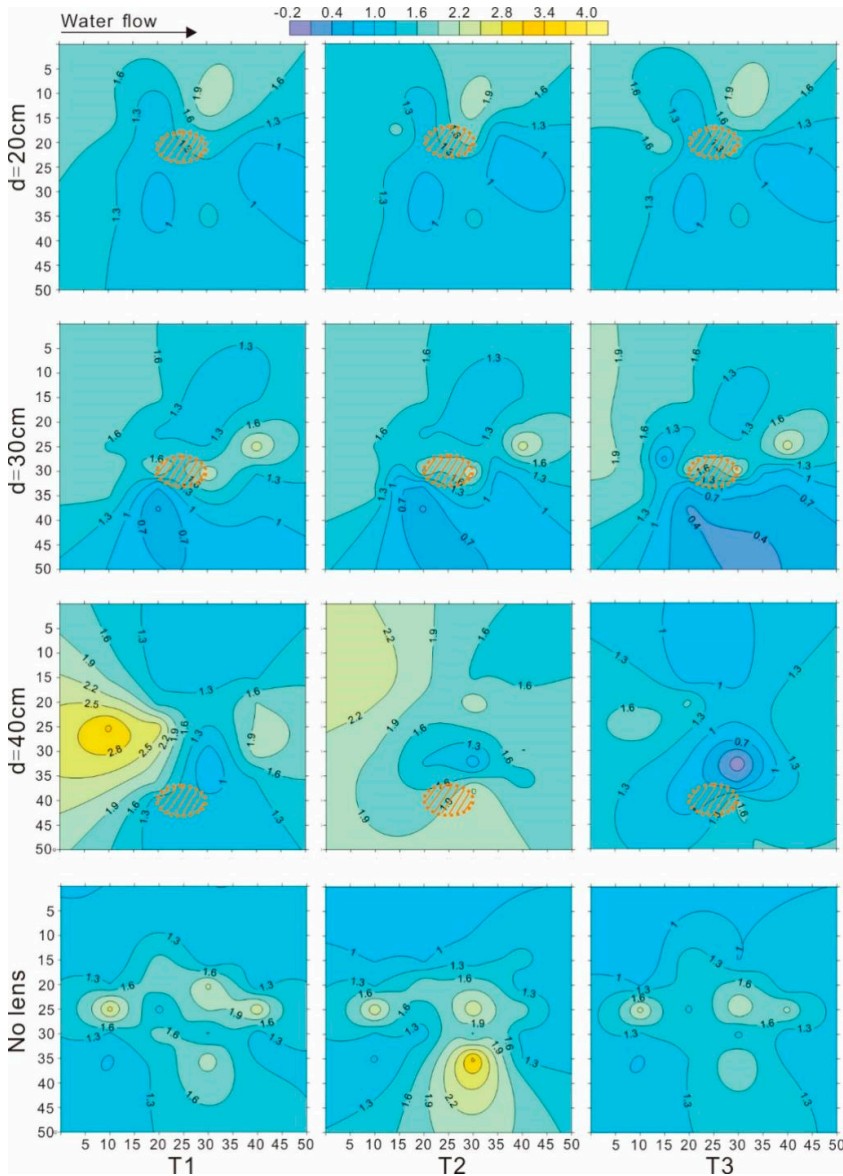

**Figure 7.** VHF fields of 12 scenarios. Brown ellipse represents clay lens, which is located at 20 cm, 30 cm, and 40 cm in depth. Columns from left to right are fields for T1, T2, and T3, respectively.

At T3, when the clay lens was located at 40 cm in depth, VHF contours were shown as concentric circles, which was similar as the stagnation zones mentioned in Gomez-Velez et al. [34]. Influenced by the surface water flow, the spatial distribution of the VHF produced an apparent distinction along the flow direction. When the clay lens was at 40 cm in depth at T1, a high-velocity zone of water flow was distributed around the area of 25 cm in depth near the upstream boundary. At T2 at 40 cm in depth, the hyporheic flow field did not form a closed high-velocity zone, but the VHF upstream was higher than that downstream.

Stonedahl et al. [67] found a strong correlation between the flow rate and the placement of high-permeability grid cells in regions of high hydraulic head gradients. In this study, the area near the upstream was prone to suffer scouring. At T1, the surface flow velocity and water depth were highest among the three points. Interestingly, the distinctions between upstream and downstream only existed when the lens was at 40 cm in depth. That is, when the lens was at 20 cm or 30 cm, this distinction could not be observed. Except for the distinction between upstream and downstream at the same points, the distinction between different points when the lens was at the same depth was also subtle.

The main differences are reflected in the depth of the clay lens, which might cause the changes in the VHF field. When the lens is in the shallow or middle layer (<30 cm), the hyporheic flow is dominated by the location of clay lens. In this situation of shallow clay lens, the influence of the surface water velocity is not significant. Thus, at the same depth, the VHF fields at each point are similar. However, while the clay lens was placed at the depth of 40 cm, the influence of the surface water velocity is obvious. It seems that the clay lens affects the hyporheic flow in local areas among the different surface water conditions.

### 4.2. Enhancement and Hindering Effects on Hyporheic Flow

For the effects of clay lens, hindering effect was more widely observed. Recently, enhancement effects were found through physical experiments and numerical simulations. Moreover, the hindering effects and enhancement effects could coexist in some cases. Su et al. [51] developed a two-dimensional dune-generated hyporheic flow model using the VS2DH model and indicated that a clay lens in streambed can hinder or enhance hyporheic flow, depending on its relative spatial location to dunes. Lu et al. [49] found the similar rule in physical model that when the clay lens was located on middle of sand dune, the effect transformed from hindering to enhancement.

In this study, hyporheic flow was embodied by the pattern of flow field, which has difficulties on distinguishing the effect of clay lens. Hence, each VHF field at different test points with a clay lens was subtracted by the blank control for corresponding sites. The new VHF fields reflected the influence of the clay lens and surface water velocity more clearly (Figure 8). Two effects of enhancement and hindering were distributed in different area and the domain of each area was no significant difference. Focus on the hindering effect, the existence of a clay lens suppressed the hyporheic flow in the surrounding area of the lens. The shape of the suppressed area also changed with the depth of clay lens. For the three test points, the VHF field at T2 was different from those of T1 and T3 to some extent, especially at a depth of 20 cm. This was indirectly affected by the surface water velocity. Our experiments graphically demonstrated that the enhancement and hindering effects of clay lens on hyporheic flow was also coexist in natural river, however it still remained challenges to quantify the range and transformation of these two effects.

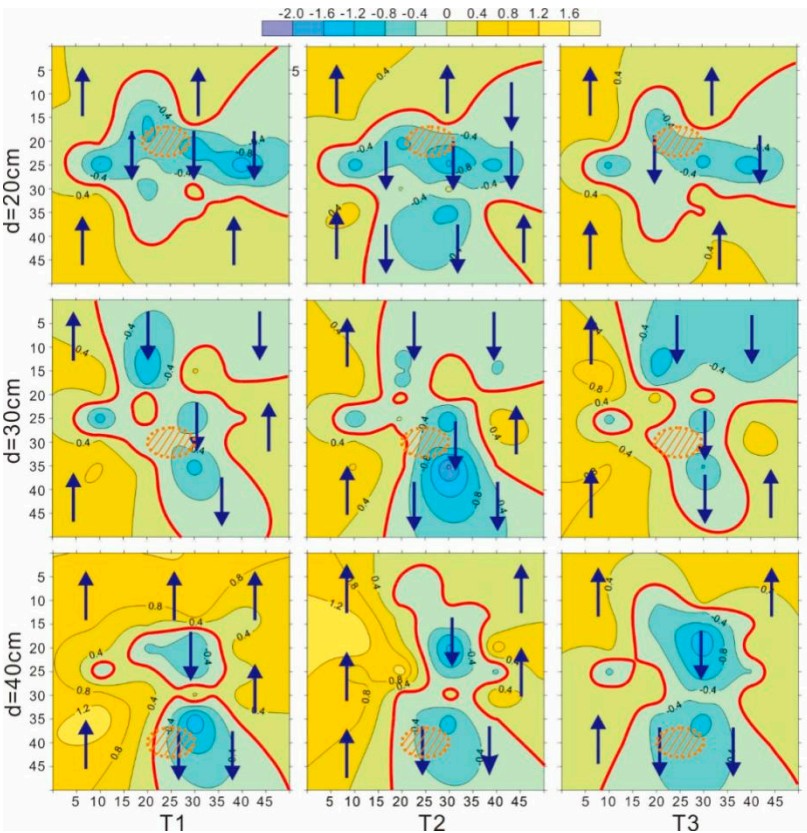

**Figure 8.** VHF field of scenarios deducting value of blank control. Directions of arrows represent signs of end results, i.e., up arrow means results are greater than 0 and the hyporheic flow was enhanced. On the other hand, down arrow means results are smaller than 0 and the hyporheic flow was hindered. Red lines are zero contours with no significant change.

## 5. Conclusions

This study used spatial temperature series data observed in a natural streambed and an artificial streambed model to study the behavior of the vertical hyporheic flux in a field test. The test in a natural streambed showed that temperature fluctuations in the streambed and air had significant differences. As the depth increased, the amplitude of the temperature curve decreased, and the temperature changes lagged in time. In general, the phase shift was less than 0, and the VHF could not be calculated by the methods of phase shift. VFLUX2 was used to calculate the VHF via six methods, which should be compared and chosen for application at specific sites. In this study, the method of amplitude ratio proposed by Hatch et al. [52] was satisfied either in robustness or in authenticity.

Field experiments in an artificial streambed showed that the vertical hyporheic flux remained stable in whole. The appearances of several extreme points resulted in significant changes to the VHF. In general, these abnormal areas of vertical hyporheic flux located in deep layers (between 25 cm and 40 cm) and the vertical flux was almost invariable.

Focusing on the heterogeneity of the sediment, the influence of a clay lens was explored by changing the depth of the clay lens. The surface water velocity was another factor that changed the characteristics of the hyporheic flow. Based on the field experiments, the relationship between the two factors was dependent with the depth of clay lens.

Comparing different depths, the influence in depth has a threshold, which was 30 to 40 cm in this study. At a value less than the threshold, the vertical hyporheic exchange is suppressed by the clay lens. When the depth exceeds the threshold, the hyporheic flux increased and appeared a high-velocity zone in the upstream bed, indicating that the hyporheic flow pattern is dominated by the surface water velocity, and the influence of the clay lens is weakened. The influence of the clay lens existed in the

local area that it decreases the local VHF and produces a "stagnation zone". This kind of abnormal velocity zone, including high-velocity zones and stagnation zones, is probably produced by the clay lens owing to rearrangement of the hydraulic conductivity distribution.

Enhancement and hindering effects were verified and distributed in different regions through the field experiments. In natural streambed, surface water velocity and depth of clay lens were the two main factors on the transformation of these two effects. However, limited by the conditions, revealing the range and transformation of two effects was still a challenge, and also the future research direction.

**Author Contributions:** All authors contributed extensively to the work presented in this paper. C.Y., C.L., W.Q. and J.L. performed the experiments and data curation; C.L. provided funding acquisition; C.Y. and C.L. wrote the paper. C.Y. and C.L. finished the review and editing.

**Funding:** This work was supported by the National Key R&D Program of China (2018YFC0407701), Natural Science Foundation of Jiangsu (BK20181035), and Fundamental Research Funds for the Central Universities (2019B10514). This study does not necessarily reflect the views of the funding agencies.

**Acknowledgments:** Many thanks to Lei Wei, Lei Qiu, Jiajie Huang and Dan Lou for their assistant in field work. We would like to thank Editage (www.editage.cn) for English language editing. This study does not necessarily reflect the views of the funding agencies.

**Conflicts of Interest:** The authors declare no conflict of interest.

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
