# Peer review of "Field Experiments of Hyporheic Flow Affected by a Clay Lens"

_water, doi:10.3390/w11081613_

Round 1
Reviewer 1 Report
Dear Authors,
The hyporetic zone is an important ecotone connecting underground and surface waters. Therefore, it is necessary to study the penetration of water from one zone to another. The title of the article gives hope for interesting results, but the content of the article gives doubts about the results and conclusions.
General thoughts:
1. The way of quoting literature differs from the guidelines of the journal. However, thanks to this, it was found that the content of the article contains items that are not on the list of literature and vice versa. The list of literature includes items that are not cited in the article.
2. Text formatting (lines 158-161, 224-229, table 1, table 2).
Detailed comments:
1. Line 93-95 - I suggest an additional hypsometric map showing the studied area.
2. Line 108 - you write the annual and monthly average precipitation and give values from 913.9 mm to 2708.4 mm. Insert a table with monthly precipitation totals or precipitation values during the tests.
3. Line 129-130 - you write that the width of sections S1 was much higher than those at S3, meanwhile Table 1 shows that it was lower (24.0 m and 27.6 m).
4. Line 199-205 - what were the dimensions of the lens. Was the lens covered with sand from all sides? It seems that the model width of 10 cm is too small to insert a lens there.
5. Line 202 - you give the dimensions of the model - I understand that these are the dimensions of "sand tank". What were the dimensions of the frame? Which side was the model in relation to the flow direction? 10 cm or 50 cm? In my opinion, placing such an obstacle in the river, even with a small flow, will cause changes in the flow and vortices will be created. Both on the sides and over the obstacle. And all this will cause temperature changes. I was expecting the model to be dug in a loose sediment.
6. Line 219-223. You write that in the second period the position of the model and the depth of the lens were constant. At what depth was the lens then?
7. In results, nowhere is marked which results are from the test period and which from the long-term period (July 20 - September 20). There is only a graph of water levels for the second period. Figures 8 and 9 show that the depth of the lens changed. Where are the results from the second period ??? Figures 4 and 7 also show the results from the first period.
8. Line 257-258. Instead of speculating about a heavy rainfall flood, give meteorological data about rainfall.
9. Section 3 Results and discussion. In this section are only results. Only Section 3.4 contains an attempt to discuss and reference the literature. I suggest dividing section 3 into two sections: 3. Results, 4. Discussion.
10. Section 4 Conclusions - in this section again you give results for the test period - line 410-411 (changing the depth of the lens), and in line 222-223 you write that for "second period, the position of the model and the depth of the lens were both constant. "
11. What is the reference of these results to other rivers with higher flow? And for rivers in other parts of the world?
Author Response
General Thoughts:
1. The way of quoting literature differs from the guidelines of the journal. However, thanks to this, it was found that the content of the article contains items that are not on the list of literature and vice versa. The list of literature includes items that are not cited in the article.
Response: Thank for your comments. We checked and revised the style of literature and the items of mistakes.
2. Text formatting (lines 158-161, 224-229, table 1, table 2).
Response: These lines and tables were reformatted according to the guidelines of the journal.
Detailed comments:
1. Line 93-95 - I suggest an additional hypsometric map showing the studied area.
Response: Agree and Changes made. The hypsometric map was shown as Figure 1.
2. Line 108 - you write the annual and monthly average precipitation and give values from 913.9 mm to 2708.4 mm. Insert a table with monthly precipitation totals or precipitation values during the tests.
Response: The precipitation values during the tests were added. See lines 106-107.
3. Line 129-130 - you write that the width of sections S1 was much higher than those at S3, meanwhile Table 1 shows that it was lower (24.0 m and 27.6 m).
Response: The flow velocity and water depth at S1 was much higher than those at S3. But the width at S1 was lower than that at S3. This mistake has been corrected.
4. Line 199-205 - what were the dimensions of the lens. Was the lens covered with sand from all sides? It seems that the model width of 10 cm is too small to insert a lens there.
Response: The dimensions of the lens have been added. In our experiment, the lens was placed horizontally that it attached the two sides of model since the height of lens was equal to the width of model. A local amplification of the lens has been added in Figure 3.
5. Line 202 - you give the dimensions of the model - I understand that these are the dimensions of "sand tank". What were the dimensions of the frame? Which side was the model in relation to the flow direction? 10 cm or 50 cm? In my opinion, placing such an obstacle in the river, even with a small flow, will cause changes in the flow and vortices will be created. Both on the sides and over the obstacle. And all this will cause temperature changes. I was expecting the model to be dug in a loose sediment.
Response: The sand tank is used to fill materials, such as sand, clay lens and sensors for experiments. As you said that the dimensions of the model are the dimensions of “sand tank”. The model is fixed with the frame in advance, which is welded by aluminum. The frame is firm enough that it could protect model from transformation or falling down. The width of frame is only ~50px, which has relatively little influence on experiment compared with the sand tank. We completely agree with your comment that bury the sand tank into the loose sediment. However, the implement is not hard in our site. The loose sediment in the study river is very thin that we could not bury the sand tank into streambed. Therefore, the sand tank was placed on the streambed directly to simulate streambed approximately and we pay more attention to the influence of the clay lens on hyporheic exchange in natural rivers. Moreover, performing an embedded experiment in a natural river is our next research plan.
6. Line 219-223. You write that in the second period the position of the model and the depth of the lens were constant. At what depth was the lens then?
Response: Due to the strong flow of flooding, the long-term model test was not finished eventually. Therefore, we only use the hydrological observations in long-term period. We are sorry about the negligence and have modified accordingly. The long-term investigation was left out in the revision.
7. In results, nowhere is marked which results are from the test period and which from the long-term period (July 20 - September 20). There is only a graph of water levels for the second period. Figures 8 and 9 show that the depth of the lens changed. Where are the results from the second period ??? Figures 4 and 7 also show the results from the first period.
Response: We observed temperature in streambed and air from July 11 to 20 (during test period) for analyzing characteristic of natural streambed temperature (section 3.1) and choosing appropriate analytic method (section 3.2). Besides, the depth of the clay lens and the position of model were changed in test period (July 10 to 20) to observe the influence of a clay lens and surface water velocity on the hyporheic exchange (section 3.3). We have deleted the contents of long-term period to avoid confusion.
8. Line 257-258. Instead of speculating about a heavy rainfall flood, give meteorological data about rainfall.
Response: Since the long-term observations of water level were not used in model test, we have deleted them.
9. Section 3 Results and discussion. In this section are only results. Only Section 3.4 contains an attempt to discuss and reference the literature. I suggest dividing section 3 into two sections: 3. Results, 4. Discussion.
Response: Agree and changes made. The new section Discussion includes the original section 3.4 and some comparisons with related references. In addition to the impacts of surface water velocity and clay lens, the enhancement and hindering effects of clay lens were discussed.
10. Section 4 Conclusions - in this section again you give results for the test period - line 410-411 (changing the depth of the lens), and in line 222-223 you write that for "second period, the position of the model and the depth of the lens were both constant. "
Response: The second period has been removed and all the results are about the model test performed during the first period.
11. What is the reference of these results to other rivers with higher flow? And for rivers in other parts of the world?
Response: This study researches the influence of a low-permeable media on the hyporheic exchange based on a sand tank to achieve quantitative control of the clay lens. Comparing with the laboratory tests (e.g. Lu et al. 2018a, Lu et al, 2018b, Fox et al. 2016) and numerical simulations (e.g. Su et al. 2018), this study reflects the characteristic of the mountain river in summer, making the hydraulic conditions more realistic. Design of the orthogonal tests and conducting tests under natural flow conditions can confirm the reasonability and confidence of results. Hence, we think that our results and discussions not only reflect the patterns of hyporheic exchange in study river, but also have a certain significance for other rivers with similar hydraulic and geological conditions. However, the conclusions should be used more cautiously in the rivers which are different in river reach in plain and the streambed made of fine material.
1. Lu C, Zhuang W, Wang S, et al. Experimental study on hyporheic flow varied by the clay lens and stream flow. Environmental Earth Sciences, 2018, 77(13):482.
2. Lu C, Yao C, Su X, et al. The Influences of a Clay Lens on the Hyporheic Exchange in a Sand Dune. Water, 2018, 10(8):826
3. Fox A , Boano F , Arnon S . Impact of losing and gaining streamflow conditions on hyporheic exchange fluxes induced by dune-shaped bed forms. Water Resources Research, 2014, 50(3):1895-1907.
4. Su X , Shu L , Lu C. Impact of a low-permeability lens on dune-induced hyporheic exchange . Hydrological Sciences Journal- Journal Des Sciences Hydrologiques, 2018, 63(5):818-835.

Reviewer 2 Report
WATER549798 “Field experiments of hyporheic flow affected by clay lens through a 2D sand tank model” C.Yao, C.Lu, W.Qin and J.Lu
This paper describes data collection and an experiment of hyporheic flux in a natural stream bed and a small tank with a clay lens. The work seems well constructed, but there is no context presented for the results. For example, what is the scale of flux compared to other sources, either in China or internationally? Are the results from the sand-box of the same orders of magnitude and therefore a reasonable, if not realistic, estimate of total flux and flux difference?
The authors suggest the odd stream height behaviour around the 17-18/09 (see Figure 4, T3 water level) where the stream level increases rapidly by 0.3m, drops about 0.6m, then back up 0.3m to the level prior to 17/09, is “probably caused by a flood”. Did the authors miss this significant event? Is there any weather data to suggest heavy rainfall? Why does the level boomerang up-down-up in the manner recorded? Why is the sudden shift on 01/08 attributed to sensor shift, but the later behaviour not?
The PDF copy supplied for review did not contain Table 4, the VFLUX parameters.
Figure 5 does not do a good job showing the flux results. Only two of eight boxes appear within the graph. The authors should consider making a graph that shows all the boxes, perhaps using a data transform so they all appear together.
Author Response
This paper describes data collection and an experiment of hyporheic flux in a natural stream bed and a small tank with a clay lens. The work seems well constructed, but there is no context presented for the results. For example, what is the scale of flux compared to other sources, either in China or internationally? Are the results from the sand-box of the same orders of magnitude and therefore a reasonable, if not realistic, estimate of total flux and flux difference?
Response: Comparing with the results of other field tests (e.g. McCallum et al. 2012, Keery et al. 2007, Lu et al. 2017) and sand-box tests (e.g. Fox et al, 2016, Lu et al. 2018), the orders of magnitude in natural river streambed ranged from -10-1 to 101 m/d, which are close to our calculations. And the results from the sand-box are below 101 m/d. Considering the difference of surface water velocity between field test and indoor tests, we think that our results from the sand-box are in reasonable range.
1. Keery, J., Binley, A., Crook, N. and Smith, J.W.N. Temporal and spatial variability of groundwater–surface water fluxes: Development and application of an analytical method using temperature time series. Journal of Hydrology 2007, 336(1–2), 1–16.
2. McCallum, A.M., Andersen, M.S., Rau, G.C. and Acworth, R.I. A 1-D analytical method for estimating surface water-groundwater interactions and effective thermal diffusivity using temperature time series. Water Resources Research 2012, 48(11), 76–78.
3. Lu, C., Chen, S., Zhang, Y., Su, X. and Chen, G. Heat tracing to determine spatial patterns of hyporheic exchange across a river transect. Hydrogeology Journal 2017, 25(6), 1633–1646.
4. Fox A , Boano F , Arnon S . Impact of losing and gaining streamflow conditions on hyporheic exchange fluxes induced by dune-shaped bed forms[J]. Water Resources Research, 2014, 50(3):1895-1907.
5. Lu C, Yao C, Su X, et al. The Influences of a Clay Lens on the Hyporheic Exchange in a Sand Dune. Water, 2018, 10(8):826
The authors suggest the odd stream height behavior around the 17-18/09 (see Figure 4, T3 water level) where the stream level increases rapidly by 0.3m, drops about 0.6m, then back up 0.3m to the level prior to 17/09, is “probably caused by a flood”. Did the authors miss this significant event? Is there any weather data to suggest heavy rainfall? Why does the level boomerang up-down-up in the manner recorded? Why is the sudden shift on 01/08 attributed to sensor shift, but the later behavior not?
Response: According to meteorological data from the nearest precipitation station in the study area, two heavy rains happened on August 31 and September 7, daily precipitation was 51.5 mm and 31 mm, respectively. Hence, we think it probably induced water level rising in September. According to the data of water level, the water level rose from 0.24 m at 12:55 on 01/08 to 0.42 m at 13:00 on 01/08, i.e. water level rose 0.18 m in 5 minutes and then did not drop. From the meteorological data there was no rain around 01/08. Therefore, the most likely cause of this phenomenon is that the position of sensor was changed. It may be washed to a deeper area.
Noted that in the new version, we leave the long-term observation out, all the analyses about the water level variation out of the test period (July 11 to 20) were removed.
The PDF copy supplied for review did not contain Table 4, the VFLUX parameters.
Response: The table has been added. See Table 4.
Figure 5 does not do a good job showing the flux results. Only two of eight boxes appear within the graph. The authors should consider making a graph that shows all the boxes, perhaps using a data transform so they all appear together.
Response: Agree and changes made. See Figure 5.

Round 2
Reviewer 1 Report
Dear Authors,
Thank you for including my comments on the article. Good luck in further work.Author Response
Thanks a lot.